# C1-C2 Rotatory Subluxation in Adults “A Narrative Review”

**DOI:** 10.3390/diagnostics12071615

**Published:** 2022-07-02

**Authors:** David C. Noriega González, Francisco Ardura Aragón, Jesús Crespo Sanjuan, Silvia Santiago Maniega, Alejandro León Andrino, Rubén García Fraile, Gregorio Labrador Hernández, Juan Calabia-Campo, Alberto Caballero-García, Alfredo Córdova-Martínez

**Affiliations:** 1Department of Surgery, Ophthalmology, Otorhinolaryngology and Physiotherapy, Faculty of Medicine, University of Valladolid, 47005 Valladolid, Spain; davidcesar.noriega@uva.es (D.C.N.G.); fardura@saludcastillayleon.es (F.A.A.); 2Department of Orthopedic, Clinic University Hospital of Valladolid, 47005 Valladolid, Spain; jcrespos@saludcastillayleon.es (J.C.S.); ssantiagom@saludcastillayleon.es (S.S.M.); aaleon@saludcastillayleon.es (A.L.A.); rgarciafra@saludcastillayleon.es (R.G.F.); glabrador@saludcastillayleon.es (G.L.H.); 3Department of Radiology, Clinic University Hospital of Valladolid, 47005 Valladolid, Spain; juancalavia@gmail.com; 4Department of Anatomy and Radiology, Health Sciences Faculty, GIR: “Physical Exercise and Aging”, University of Valladolid, Campus Universitario “Los Pajaritos”, 42004 Soria, Spain; alberto.caballero@uva.es; 5Department of Biochemistry, Molecular Biology and Physiology, Health Sciences Faculty, GIR: “Physical Exercise and Aging”, University of Valladolid, Campus Universitario “Los Pajaritos”, 42004 Soria, Spain

**Keywords:** atlantoaxial joint, atlantoaxial subluxation, adults, C2, C1, cervical trauma

## Abstract

The atlantoaxial joint C2 (axis) with the anterior arch of C1 (atlas) allows 50% of cervical lateral rotation. It is responsible for precise and important movements that allow us to perform precise actions, both in normal and working life. Due to low incidence in adults, this condition often goes undiagnosed, or the diagnosis is delayed and the outcome is worse. An early diagnosis and treatment are essential to ensure satisfactory neurological and functional outcomes. The aim of this review is to analyze C1-C2 rotatory subluxation in adults, given its rarity. The time between injury and reduction is key, as it is directly related to prognosis and the severity of the treatment options. Due to low incidence in adults, this condition often goes undiagnosed, or the diagnosis is delayed as a lot of cases are not related to a clear trauma, with a poor prognosis just because of the late diagnosis and the outcome is worse. The correct approach and treatment of atlantoaxial dislocation requires a careful study of the radiological findings to decide the direction and plane of the dislocation, and the search for associated skeletal anomalies.

## 1. Introduction

Non-traumatic rotatory subluxation of C1 over C2 (atlantoaxial rotatory subluxation) (ARAS) is a rare condition. Atloaxial (atlantoaxial) subluxation is a disorder of the cervical spine, at the level of the C1 and C2 vertebrae, which causes impaired neck rotation because the anterior facet of C1 is fixed to the facet of C2. Moreover, hyperemia is the most accepted pathogenic model cause. This is a situation following infections or surgical trauma leading to decalcification of the anterior arch of the atlas and laxity of the anterior transverse ligament between the atlas and axis [1].

The atlantoaxial joint C2 (axis) with the anterior arch of C1 (atlas) allows 50–60% of cervical lateral rotation [2]. In this joint, the superior articular facets of the atlas are concave in both directions and oval; they are oriented upwards and inwards, to accommodate the occipital condyles, whose articular faces are oval and convex caudally [2]. The medial atlantoaxial joint is classified as a trochoid-type synovial diarthrosis. As a whole, it presents three axes of motion with three degrees of freedom, although its wide range of rotational motion stands out. There are no intervertebral discs between the occipital and atlas and between the atlas and axis. This specific design of the C1-C2 joint is responsible for precise and important movements that allow us to perform precise actions, both in normal and working life.

This joint performs angular motion (flexion and extension), rotation (right and left), and linear motion and translation (anterior and posterior, right and left). When the odontoid is normal, the anterior motion of the axis vertebra (C2) is restricted by the anterior arch of the atlas (C1). Posterior motion is limited by the cruciate ligament, largely by the transverse ligament [3,4].

It is important to recognize that this pathology is a clinical-surgical emergency, potentially causing severe neural damage, long-term sequelae, and even death if not treated immediately. The time between injury and reduction is key, as it is directly related to the capacity (facility) to reduce the joint and the prognosis [5,6]. Surgical treatment should be immediate, but conservative treatment lowers the prognosis if the diagnostic delay is longer than 1 month [5,6]. When the diagnosis is delayed, reduction of the dislocated atlantoaxial joint can be extremely difficult, requiring several weeks of skull traction without over distraction in unstable lesions [5,6]. Due to low incidence in adults, this condition often goes undiagnosed, or the diagnosis is delayed and the outcome is worse. In patients with head and/or spine trauma, the delay between injury and reduction predisposes to recurrence of this condition and the inability to heal after non-surgical treatment with consequent loss of upper cervical spine mobility.

### Objective

The aim of this review is to analyze C1-C2 rotatory subluxation in adults, given its rarity. This joint is responsible for precise and important movements that allow us to perform precise actions, both in normal and working life. Likewise, this pathology often goes unnoticed and undiagnosed. The time between injury and reduction is key, as it is directly related to the prognosis and severity of treatment options. The correct approach and treatment of atlantoaxial dislocation requires a careful study of the radiological findings to decide the direction and plane of the dislocation, and the search for associated skeletal anomalies.

## 2. Materials and Methods

A structured search was conducted in the SCOPUS, Medline (PubMed), and Web of Science (WOS) databases. The search used the following keywords “C1-C2, atlantoaxoidea, atlantoaxial, AND rotatory AND (instability OR Subluxation OR Dislocation)” As inclusion criteria, we have established: human, adult, and English. All found titles and abstracts were separated to identify duplicates and possible missing studies. We first selected the articles based on what we found in the abstracts, which led to a more exhaustive selection by reading the selected articles. We have to take into account that many of them are very similar and do not contribute more than another selected article. The bibliographic search was conducted according to the following guidelines [7]. The “Full search strategy” is presented in Figure 1.

## 3. Etiology

C1-C2 rotatory subluxation (Grisel syndrome) was described predominantly in the pediatric population, although due to its possible causes it is not infrequent in adults. It was first described by Bell in 1830 as a consequence of a syphilitic ulceration of the pharynx [7,8,9].

However, despite the fact that this syndrome has been known for more than a century, its pathophysiology is still not well understood. Several mechanisms have been suggested that could be differentiated into traumatic and or non-traumatic. One of the mechanisms is that produced by “blow” (whiplash) which leads to spasm and subsequent subluxation [7,10]. On the other hand, anatomical studies show the existence of a periodontoidal vascular complex, which drains the posterior superior pharyngeal region [11].

Other authors suggest that it is due to “cervical lymphadenitis” caused by a nasopharyngeal infection that can provoke a spastic contraction of the suboccipital and paravertebral muscles, causing torticollis [12]. We could suppose that any inflammatory situation in the head or neck could be an etiological factor. Cases derived from surgical procedures due to manipulation of the cervical spine in the process of anesthesia or during the surgical intervention itself have also been described [13]. In addition, cases of juvenile idiopathic arthritis have been reported [14]. It may also be associated with congenital bone anomalies of the occipital–atlas–axis complex [15]. It has even been observed with minimal trauma due to the application of an orthopedic device or even the removal of a cast [16]. The instability of these joints is accompanied by compression of cervicomedullary neural structures resulting in neurological disability. Regardless of the etiopathogenic mechanism, discomfort of the atlantoaxial complex leading to muscle spasms and neurological involvement is common.

Although the pathogenesis is not clear, Barcelos et al. [17] argue that the inflammation and hyperemia typical of the pathogenic processes cause an increase in the laxity of the anterior transverse ligament, with the consequent subluxation of the joint.

### 3.1. Biomechanics of C1-C2

The cervical spine can rotate up to 90 degrees, with approximately 50–60% of the rotation of the atlantoaxial joint, a movement that is aided by the horizontal orientation of the C1-C2 facet joints, allowing further rotation without compromising bony stability [2,18,19]. This mobility is promoted by the ligamentous structure. The transverse ligament keeps the odontoid opposite the anterior arch of C1 and prevents excessive anterior movement of the atlas on axis during movement. The paired alar ligaments running from the tip of the odontoid to the right and left occipital condyles prevent anterior subluxation and work with the C1-C2 capsular ligaments to resist excessive rotational motion [18,19]. As published by Roche et al. [18], hypermobility of C1 over C2 occurs when atlantoaxial rotation is greater than 56 degrees or a right–left difference greater than 8 degrees. Hypomobility is diagnosed when the angle is less than 28 degrees [19].

### 3.2. Classification

The different classifications are based on the various etiological factors and diagnostic tools used.
(a)Traumatic basis (Table 1)(b)Based on the image

The most commonly used classification is Fielding and Hawkins [20,21] (Figure 2). Type I consists of a unilateral anterior rotation of an atlas mass that pivots around the odontoid while maintaining the integrity of the transverse ligament of the atlas [20,21,22]. As indicated by Ng et al. [23], rupture of the alar ligament may be a negative prognostic indicator for the success of non-surgical treatment of type I atlantoaxial rotator subluxation. Type II, in which there is also a unilateral anterior displacement of an atlas mass, is performed pivoting on the contralateral atloaxoid joint resulting in an atloaxoid separation of up to 5 mm. The transverse ligament of the atlas may, in this case, be deficient. Type III occurs when there is anterior subluxation of both lateral masses of the atlas. The atloaxial separation may exceed 5 mm, which presupposes a totally incompetent transverse ligament. The rare type IV cases represent a uni or bilateral posterior subluxation of the atlas masses, usually in coexistence with an epiphysiolysis or with an agenesis of the odontoid.

Other authors such as McGuire et al. [24] have proposed another classification based on dynamic computed tomography (DCT), dynamic CT. According to this, three stages are distinguished: (a) stage 0, the DCT was normal; (b) stage 1, there is less than 15° difference between the rotation of C1 and C2, but C1 crossed the midline; (c) stage 2, the C1-C2 movement is fixed (Figure 3).

For their part, Pang and Li et al. [16,19], also using DBT, established C1-C2 motion curves, whose information was compared with normal control subjects: (a) group 1, the C1-C2 angle corrected to the maximum C1-C2 angle decreased by 20% from the presentation position; (b) group 2, C1-C2 angle reduction of more than 20%; (c) group 3, C1 rotated to the opposite side, but dragged C2 with it, so that C2 rotated an abnormal amount to the corrected side; (d) group 4 showed C1-C2 motion within normal parameters; (e) group 5, gray area in which there was only slight abnormal coupling of C1-C2 motion.

Other authors [25,26] using three-dimensional computed tomography (TTC) classified atlantoaxial rotational subluxation (RAS) according to the lateral tilt of the atlas on its axis and the presence of C2 deformity. According to them, they made the following scale:

Grade I patients who presented torticollis in 8 weeks and had no facet deformities.

Grade II patients who presented torticollis at 3 months and had facet deformity of less than 20° of C1 inclination.

Grade III patients who presented with a facet deformity greater than 20° of C1° inclination.

## 4. Clinical (Symptomatology)

The most frequent symptom is cervical pain accompanied by torticollis with neck stiffness. The pain may radiate by nerve compression to the occipital region, face, and ears. Patients present with painful torticollis; in some cases, they mention or suffer from upper airway infections. On physical examination, muscle spasm of the neck muscles is usually observed, with the impossibility of voluntary correction of torticollis [26].

It is common to find the patient with the head rotated to one side, being tilted to the other (cock-robin position). The patient is unable to rotate the head to the other side and is usually associated with an intense spasm of the sternocleidomastoid muscle medial to the rotation. Any voluntary or forced attempt to restore the normal position of the head causes severe cervical pain and is usually unsuccessful. Neurological symptoms are almost never present due to the looseness of the vertebral canal at this level [27].

Neurological involvement is unusual in the atraumatic presentation unless there is an anterior or posterior displacement of the atlas. However, in some cases and depending on its etiology, for example, in cases derived from rheumatoid arthritis, neurological symptoms are present, which can be very varied, and occur less frequently than pain. We can find signs of spinal cord compression at the cervical level that is accompanied by broad-based spastic gait, balance disturbances, and decreased fine motor skills in the hands. Hyperreflexia, Babinski’s and Hoffmann’s signs, clonus, and inverted brachioradial reflex, among others, are also described. The sign of “L‘Hermitte” which is the sensation of electricity in the region of the dorsum and extremities, as a consequence because of neck movement in an unstable spine, is another finding that suggests the presence of spinal cord compression. In the long term, urinary retention and varying degrees of muscle weakness in the extremities may also be observed [27].

Symptoms associated with vertebrobasilar insufficiency due to atlantoaxial impaction include: lipothymias, vertigo, dysphagia, seizures, tinnitus, balance disturbances, dysarthria, nystagmus, or visual deficits [27].

## 5. Diagnosis

Although the diagnosis is eminently clinical, the fundamental diagnostic tool is the radiological study [16,21,25,26,27]. However, in the first place, and given that many of the cases are accompanied by inflammation of the head and neck (regardless of whether it is infectious in nature or not), tests indicative of possible inflammatory disease should be performed. A complete blood count should be performed, with an assessment of the erythrocyte sedimentation rate (ESR) and C-reactive protein (CRP) level [27]. From a radiological point of view, imaging may be useful to aid in staging and to rule out trauma or inflammatory processes. X-rays play a key role [27,28].

### 5.1. Simple X-ray

Anteroposterior (AP) X-ray with the mouth open may show displacement of the odontoid process or lateral overlap [28]. However, the head position and limited range of motion associated with torticollis can make cervical spine radiographs inaccurate and difficult to interpret [29]. In lateral radiography, head tilt can obscure the normal anatomy and landmarks of the vertebrae, thus requiring expertise and good orientation in their positions.

### 5.2. Computed Tomography (CT)

This has been a widely used technique, and even based on different classifications have been made as we have indicated above [16,19,21,24,30]. Volumetric CT shows the rotation of C1 over C2 with exposure of part of the superior articular facet of C2, indicating rotatory subluxation (Figure 4 and Figure 5).

Recently, the diagnostic value of CT has been questioned, especially in acute phases. In this sense, Alanay et al. [31] indicated that it had low reliability and reproducibility and advised against its routine use. In the same way, Hicazi et al. [32] also did not observe movement abnormalities 4 days after the onset of symptoms. This could be due to the fact that patients with acute torticollis probably have fewer abnormalities.

### 5.3. Magnetic Resonance Imaging (MRI)

MRI is currently the technique of choice because it can be performed without the risk of ionizing radiation, and it can detect edema in the C1-C2 complex (Figure 6). MRI can also be useful if there are other etiological factors such as a tumor or infection [33]. In addition, MRI is very useful for the identification of cervical spinal cord compression by defining the bony or inflammatory elements causing the stenosis.

The use of dynamic MRI has also been proposed for evaluation since patients with a spinal cord diameter of less than 6 mm in flexion are at risk of neurological deficit.

To summarize, we could say that plain radiography in lateral projection is essential to assess signs of instability. Thus, an anterior displacement or an increase in the interspinous space may indicate a posterior lesion of the subaxial spine. A better assessment of the bone structure is provided by CT images, which should be taken systematically not only for diagnostic purposes but also to predict the correct placement of the screws on the lateral masses. However, it is the MRI study in which images provide the most data to demonstrate the lesion of the posterior tension band [34].

## 6. Treatment

The primary goal of treatment in patients with non-traumatic rotatory subluxation of C1 over C2 (SRAA) is to relieve pain and prevent the development of neurological deficits. The decision to adopt a surgical approach is based on the stability of the joint, its relocation, and the involvement of the transverse alar ligaments [23,35]. It should be noted that in untreated cases pathological adhesion may occur between C1 and C2, which increases with the persistence of abnormal dynamics, probably due to contracture of the periarticular soft tissues and chronic adhesive changes on the articular surfaces [36].

### 6.1. Conservative Treatment

Conservative treatment includes the use of analgesic and anti-inflammatory drugs, muscle relaxants, physical therapy techniques, and orthoses. In fact, there are authors [29] who have reported the resolution of ARAS with medication alone. However, this is usually supplemented with some form of passive cervical manipulation. Within physical therapies, options for realigning the neck in a reduced position include the use of cervical collars, halter or skeletal traction, or halo immobilization. In the absence of fracture or neurological compromise, non-surgical treatment is a viable option before resorting to surgery.

Fath et al. [35] have proposed a therapeutic approach according to the Fielding–Hawkins classification, which we reproduce in Table 2 below.

In general, types I and II are reduced and return to functional normality in less than a week with simple therapeutic measures (soft orthoses, analgesics, and muscle relaxants). When they persist for more than a week, manipulations or tractions would be necessary for their reduction, followed by the placement of containment systems such as molded minervas or, according to some authors, craniothoracic halos. Recurrent cases, not reducible by manipulation or after three weeks of orthotic treatment, may require surgical treatment with subsequent instrumented fixations. Cases with type III or IV dislocations will require surgical treatment with posterior instrumented fixations. In the case of type III fractures, their trajectory extends from the base of the odontoid to the body of C2 and treatment is either clinical or surgical, depending on stability/instability, leading to the use of an internal or external fixation method [35].

There are different protocols for the establishment of an orthosis until the clinical resolution of torticollis. Several authors are of the opinion that the recommended time of orthosis use would range from 1 week to 6 months [24,30,32,36,37]. However, Landi et al. [33] indicated that to establish the duration of the orthosis it was necessary to perform an MRI control.

Regarding skeletal traction, there are authors [26] who had no relapses after its use, followed by an orthosis. However, these same authors, also with skeletal traction, in patients with symptoms of more than 8 weeks duration, did not obtain the reduction in ARAS or had a recurrence after reduction.

This same research group [38,39], in patients with chronic RSAA, performed treatment using closed manipulation and halo vest devices. The authors monitored treatment time by 3D CT scanning and observed that dysplasia was reversed after immobilization for 2.8 months with the halo. Spinal distraction with halo traction is useful for the treatment of selected complex cervical and high thoracic deformities [40,41]. As indicated by Yeung and Feng, it is possible that in most cases the use of the halo could achieve the restoration of normal cervical alignment [41]. In this way, Kia et al. [42] closed reduction should be attempted; however, the chronicity of the injury may place it at a high risk of recurrence if arthrodesis is not performed. Kia et al. [42] have indicated that closed reduction should be attempted, although the chronicity of the lesion may pose a high risk of recurrence if arthrodesis is not performed.

From the orthopedic point of view, different techniques and cranio-cervical fixation systems have been proposed. From the lowest to the highest immobilization capacity, they are compiled in different treatises [43,44] and from which we extract the most used techniques:Philadelphia or Aspen-type rigid collar. It restricts cervical mobility partially, and above all in flexion-extension, being less effective for rotation and lateral flexion.SOMI (sternal occipital mandibular immobilization): is a system that allows immobilization in extension, flexion, or neutral position. The support is by means of anterior and posterior bars that are adjustable for the flexion-extension position and the support base extends over the scapulae and thorax.Cranial-thoracic halo with mentonian and occipital support. The support is by means of anterior and posterior bars that are adjustable for the flexion-extension position and the support base extends over the scapulae and thorax.Cervicothoracic Minerva. Provides frontal, occipital and mentonian support.

### 6.2. Surgical Treatment

Undoubtedly, the type of treatment required is based on the duration of symptoms, i.e., presence of instability, neurological deficit, and intractable cervical pain, which will be the most important factor in predicting the treatment required [16,45,46]. The cervical spine allows head movement in all three dimensions of space, hence cervical fixation systems are essential for both reduction and immobilization, as well as for choosing the treatment. However, there is no clear way to determine the acute consideration of ARAS, although these cited authors indicate that the acute phase is within 1 month from the onset of symptoms.

The problem arises in patients who, despite having radiological instability, do not have neurological alterations or significant pain. To resolve this situation, authors such as Florensa [47] propose that early surgical intervention can help prevent progression of the disease.

Surgical indications include: the presence of instability, neurological deficit, and intractable cervical pain. When surgery is needed, fixation of C1-C2 [44,47,48,49] using lateral masses and screws between the articular parts generates great stability in all planes of the atlantoaxial joint complex. This invasive therapy is recommended only when conservative treatments do not lead to remission. Therefore, the treatment of C1-C2 instability requires in many cases surgical stabilization by means of arthrodesis techniques. Table 3 below lists some of the surgical techniques used in the treatment of AARS. In a recent review, Huang et al. [48] have listed the different techniques used.

Therefore, the assessment of clinical instability of the spine is key. In this sense, White and Panjabi, based on anatomical (radiological), biomechanical and clinical studies, created the so-called “White and Panjabi clinical checklist”. These authors conclude that the stability of the anterior spine depends fundamentally on the anterior and posterior cervical ligaments, while the posterior spine depends on the articular capsules of the articular facets [3].

There are other classifications that also provide guidance on instability and the procedures to be followed depending on the mechanisms of injury. For example, the Allen–Ferguson classification, based on the cervical position at the time of injury; or the SLIC classification (Subaxial Injury Classification and Severity Scale) based on morphology, disc-ligamentous status, and neurological involvement [43,49,50].

#### Arthrodesis

Arthrodesis of the C1-C2 vertebrae is one of the techniques of choice when non-surgical treatment fails. The osteosynthesis devices must create multidirectional stabilization by short segmental fixation, leaving as many mobile segments free as possible [51,52,53,54,55,56]. In short, cervical spine fixation makes it possible to restore stability or correct the alignment of the vertebrae. The main risk of spinal fixation is associated with the neurological lesion that can occur when inserting the arthrodesis material, for which surgeons use radiological methods to continuously control the location of the implanted material at all times. In the development of this surgical technique, different methods have been described, all of which have the same objective. In addition, as the instability of the cervical spine is the fundamental issue, the differences in the contribution to the range of motion of C1 and C2 must be taken into consideration, which leads to different criteria being established according to the different types of associated fractures.

There are many surgical techniques described for C1-C2 fixation, including wire and bone graft fixation, transpedicular screws, and combinations of these techniques. Since it would be very extensive to describe each of the techniques according to the original cause of the ARAS, we will only mention those in which there is the greatest consensus [57,58,59,60,61,62,63]. Among these techniques we will mention some of them:

The first reference of stabilization of C1-C2, was the one technique presented by Mixter and Osgood [63]. In this technique, they passed a braided silk under the posterior arch of C1 to subsequently tie it to the spinous process of C2.
Magerl’s technique indicated for posterior cervical spine fixation.Harms and Melcher [64]: atlantoaxial stabilization technique by individual fixation of the lateral mass of C1 and the pedicle of C2 with minipoliaxial screws and rods.Gallie fusion technique. Gallie, in 1939, described the first posterior wiring technique for C1-C2 fixation, this technique used a single, central wire displaced sublaminarly under the posterior arch of C1 and around the spinous process of C2 that fixed in place a bone graft.The modified Gallie technique by Volker Sonntag. Sonntag’s technique improves the rotational stability of the Gallie fusion technique while avoiding the bilateral C1-C2 sublaminar wire passage of the Brooks–Jenkins technique [65,66].Brooks and Jenkins introduced the fusion technique using two trapezoidal bone grafts between the posterior arch of C1 and the laminae of C2 secured with two wires that were placed sublaminarly [66].Dickmann and Sonntag technique, consisting of a modification of the previous techniques. In this case, the sublaminar wires were eliminated at C2, and the spinous process of C2 was used as the anchor point. Intraoperatively, the C1-C2 interlaminar space was widened.Goel and Laheri technique [67,68] developed an atlantoaxial fusion technique that was later popularized by Jurgen Harms and Robert Melchor in 2001 [61]. The technique consists of using polyaxial screws in the lateral mass of C1 and in the pedicle of C2 and rods. Once the screws have been placed, reduction in the dislocation is achieved, and the rods are then placed.Interlaminar clamp technique provides a fusion similar to the Brooks–Jenkins method, but without the disadvantage of sublaminar wires.Halifax technique uses a double hook and screw that stabilizes the laminae of C1 and C2 bilaterally and secures bilateral interlaminar bone grafts [69].

## 7. Discussion

The atlantoaxial joint and its concomitant ligaments provide significant rotational and biomechanical stability to the cervical spine. Atlantoaxial subluxation is a rare pathological situation of instability in adults. Isolated traumatic rotatory subluxation of the atlantoaxial joint without fracture is a rare injury, without consensus for management in the adult population. Its treatment is in two ways: conservative and surgery. Based on the production procedure of the C1-C2 subluxation, new classifications have also been proposed in order to better clarify the therapeutic possibilities [70].

However, when it comes to a broad discussion of this pathological situation, we find that most of the publications correspond to a clinical case or review (Table 4). We intend to make a brief discussion of the situation, for which we have prepared a summary table with the largest and most important manuscripts, which may allow us to draw conclusions.

According to the scientific literature, C1-C2 instability (atlantoaxial subluxation) is a problem with multiple etiologies [40,70,71,72,73,74], which, due to the mild neurological symptoms [75] in the initial phase, can be difficult to diagnose by the complexity of the interpretation of simple imaging tests. This fact generates delays in diagnosis and will compromise the capacity for reduction and recurrence of the deformity [40]. In this initial phase, clinical abnormalities in cervical spine deformity and limited mobility [75], should lead us to investigate the possibility of C1-C2 instability even in cases where there is no obvious trauma or cause [73,75].

The speed of diagnosis has an important influence on the future of the joint and, therefore, on the functionality of the cervical spine, because this will determine the possibility of a bloodless reduction [40,71,73] or surgical treatment, as have proposed by Isik et al. [72]. In agreement with Graziano et al. [40], treatment is conditioned by the speed of diagnosis. In addition, this is conditioned by the possible recurrences associated with it, as Sinigaglia et al. have indicated [71]. These authors explain the difference between treatment in the acute or subacute phase, with less therapeutic aggressiveness and better long-term results. This behavior can also be seen in the cases of Graziano et al. [40].

In short, and in agreement with the aforementioned authors, early diagnosis is a determining factor in therapeutic aggressiveness and long-term clinical and functional outcome.

## 8. Conclusions

The C1-C2 rotatory subluxation has different and varied etiologies that determine the therapeutic approach to be followed. Common in all of them is pain, inflammation, torticollis, and in more advanced cases, neurological complications. Other aspects not open to discussion are the diagnostic elements, i.e., symptomatology and imaging studies. A late diagnosis can have serious consequences. Systematic procedures and a clear methodology in radiological evaluation will avoid diagnostic delay. However, when studying and recognizing the different surgical treatment options, there are numerous techniques available, all of them leading to stabilization. However, there is a relevant fact: the techniques have not evolved at all since they were introduced (starting in 1930) up to the present day and until today. Perhaps with the passage of time and with the help of artificial intelligence, we will be able to have clear working schemes for the application of surgical techniques. Hopefully, in the future, and with the help of artificial intelligence, we will be able to have clear protocols for the application of surgical techniques.

Undoubtedly, improving the results is key to early diagnosis, since this would preserve the normal biomechanics of the spine. By this, adequate management in the Emergency Department is very important. The problem should never be underestimated, and it is necessary to have a good image to confirm it.

## Figures and Tables

**Figure 1 diagnostics-12-01615-f001:**
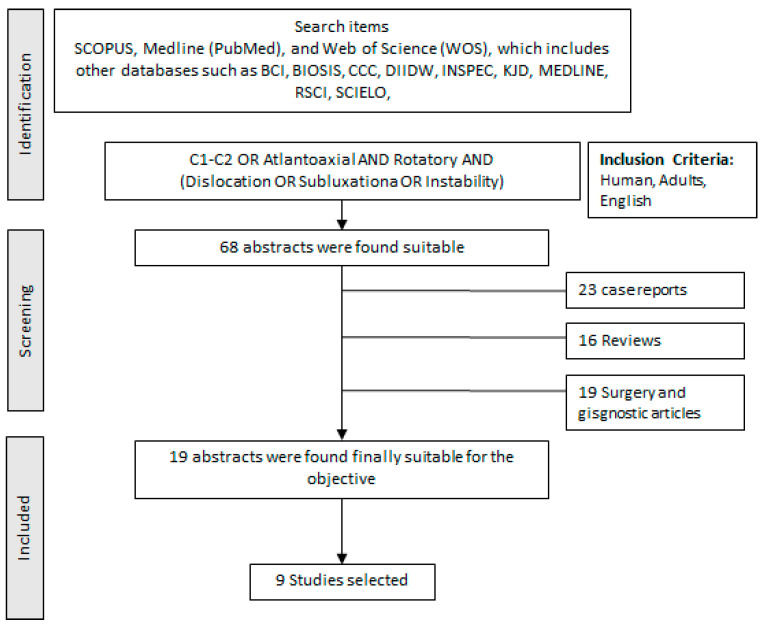
Full search strategy to develop the systematic and after narrative review.

**Figure 2 diagnostics-12-01615-f002:**
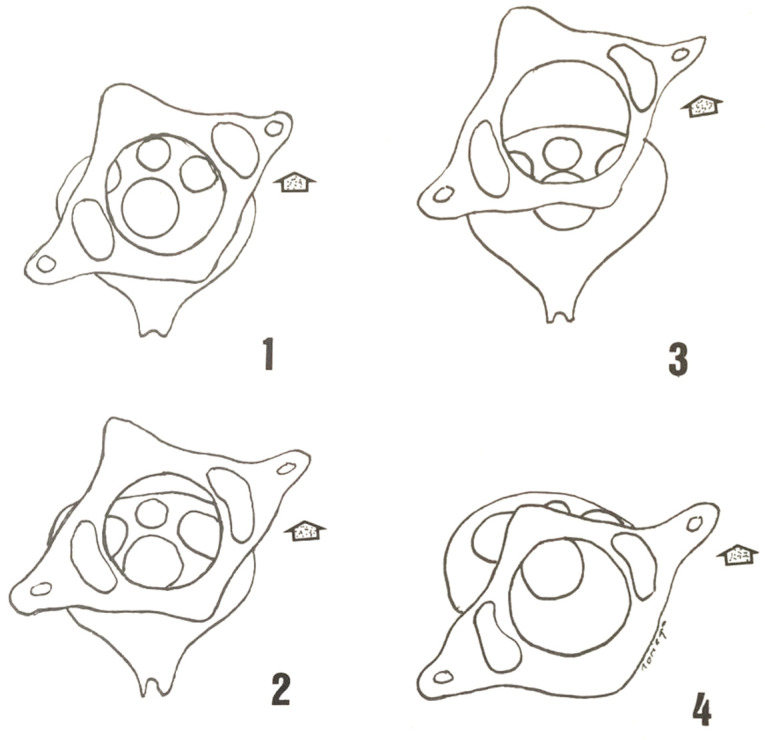
Fielding and Hawkins [20] classification of C1-C2 rotatory subluxation.

**Figure 3 diagnostics-12-01615-f003:**
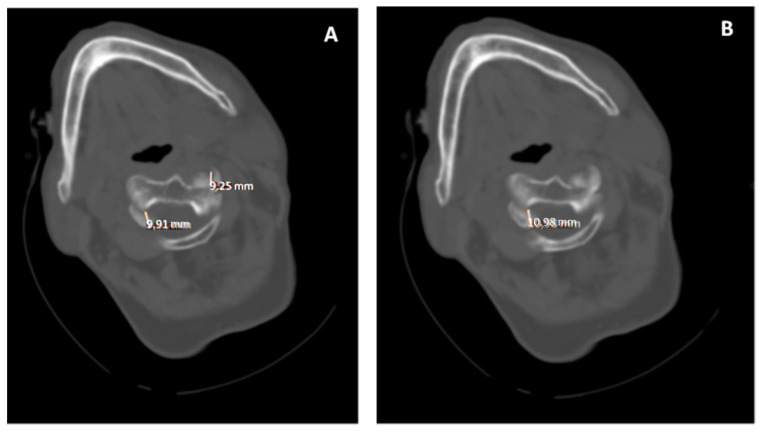
(**A**,**B**). Computed axial tomography, axial slices C1-C2. Traumatic rotatory dislocation C1-C2. There is an asymmetry in the atlas-axis distance with a greater distance from the odontoid towards the atlas on the left side.

**Figure 4 diagnostics-12-01615-f004:**
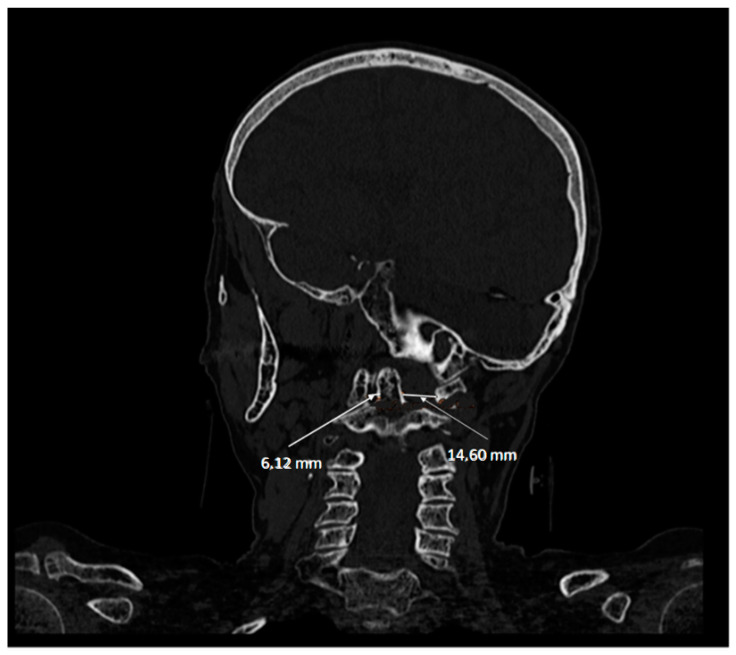
Computerized axial tomography, sagittal section. The asymmetry of the left and right articular facets can be seen.

**Figure 5 diagnostics-12-01615-f005:**
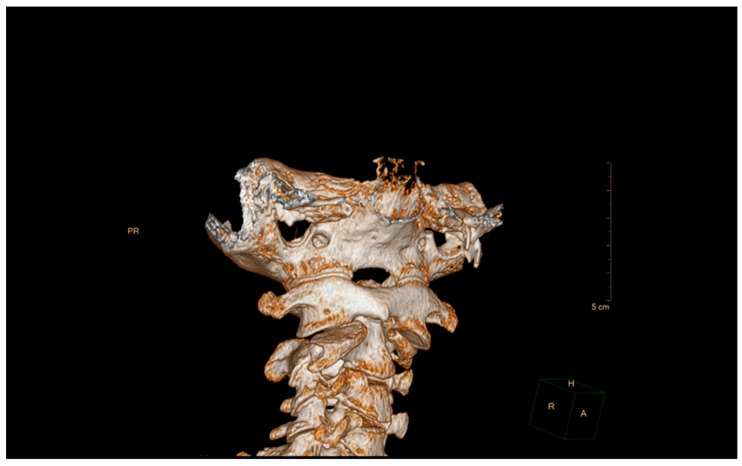
Computed axial tomography, 3D reconstruction.

**Figure 6 diagnostics-12-01615-f006:**
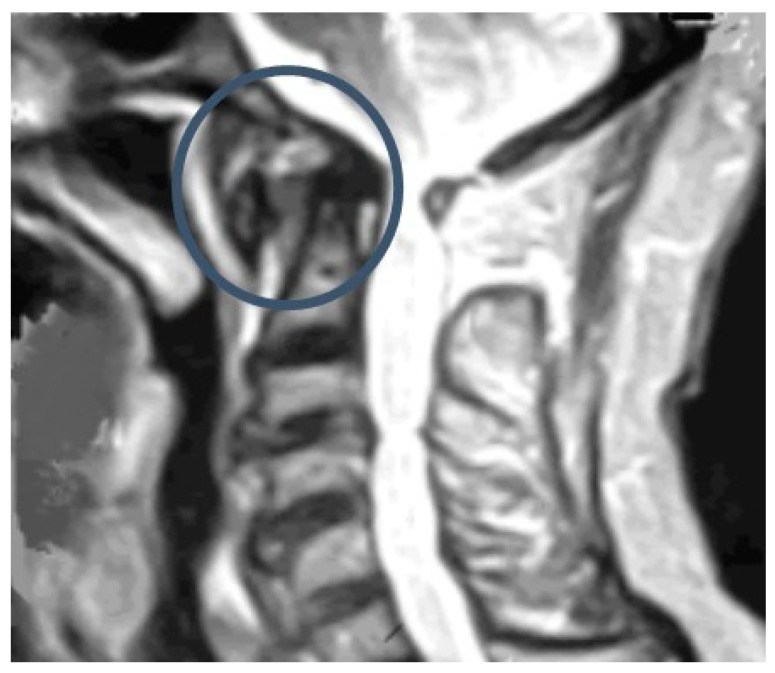
Subluxation C1-C2 by MRI.

**Table 1 diagnostics-12-01615-t001:** Traumatic causes of instability of the upper cervical spine (C1-C2):

- Occipitoatloid dislocation: is produced by the total rupture of the ligaments and joint capsules that join the atlas and odontoid to the occipital bone. - Atloaxial (atlantoaxial) rotatory dislocation: it may be caused by minor trauma, involving sagittal rotation of the trunk, although it is also found in severe trauma. - Fractures of the atlas: account for 5–15% of all cervical spine fractures and 1–3% of all spine fractures. Axis fractures are associated in half of all cases. Axis fractures: account for 20% of all cervical fractures.

**Table 2 diagnostics-12-01615-t002:** Therapeutic strategy according to the Fielding–Hawkins classification.

Fielding–Hawkins Classification	Description	Therapeutic Strategy
Type I	Pure rotation of the atlas in relation to the axis without anterior displacement	Conservative treatment: soft collar, non-steroidal anti-inflammatory drugs (NSAIDs), muscle relaxants.
Type II	Rotated atlas with anterior displacement of 3 to 5 mm	Conservative treatment, reduction, Philadelphia brace
Type III	Atlas rotated with anterior displacement greater than 5 mm	Surgical treatment, open reduction, fusion of C1-C2

**Table 3 diagnostics-12-01615-t003:** Surgical techniques in traumatic pathology of the RAAHS.

- Transarticular arthrodesis - Anterior odontoid screw fixation - Interlaminar fixation - Atlantoaxial fixation - Endoscopic techniques - Mixed techniques

**Table 4 diagnostics-12-01615-t004:** More extensive and more relevant studies.

Author.	Etiology	Mainly Symptom	Complications	Treatment
Xu et al. [71]	Varied	Varied	No	Surgery
Sinigaglia et al. [72]	Traumatic	Pain Deformity	Residual pain at follow-up	Halo
Isik et al. [73]	Rheumatoid arthritis	Pain Myelopathy	No	Surgery
Kim et al. [74]	Iatrogenic-postsurgery	Pain Deformity	No	Conservative Traction Surgery
Graziano et al. [41]	Rheumatoid arthritis Traumatic Ankylosing spondylitis	Deformity Renal and cardiopulmonary problems	Cast sores Pin infection	External fixator (50% recurrence)

## Data Availability

Not applicable.

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
