# Peer review of "C1-C2 Rotatory Subluxation in Adults “A Narrative Review”"

_diagnostics, 2022, doi:10.3390/diagnostics12071615_

Round 1

Reviewer 1 Report

Dear Authors,

The review in question is interesting, as it analyzes a frequently underestimated problem in musculoskeletal clinical practice.

My suggestion is to exploit what has been done to follow up on the problem raised by you in your conclusions, inherent the immobility in terms of evolution of the therapeutic approaches to the problem under consideration: what could be, more specifically, the viable paths at diagnostic and therapeutic to improve surgical and, above all, conservative management? What should any experimental studies on the subject foresee and test?

Finally I would like to point out a small doubt about the linguistic rendering of the sentence present in line 59-60 "The time between injury and reduction is key, as it is directly related to the capacity to reduce the joint and the prognosis.". What is meant by "the capacity tu reduce the joint"?

Renewing my congratulations for the paper produced, I wish you good work.

Author Response

The review in question is interesting, as it analyzes a frequently underestimated problem in musculoskeletal clinical practice.

My suggestion is to exploit what has been done to follow up on the problem raised by you in your conclusions, inherent the immobility in terms of evolution of the therapeutic approaches to the problem under consideration: what could be, more specifically, the viable paths at diagnostic and therapeutic to improve surgical and, above all, conservative management? What should any experimental studies on the subject foresee and test?

Thank you very much for your appreciation. We put the aspect that you suggest in the conclusion (in blue) to make the subject clearer.

ANSWER

“Undoubtedly, to improve the results is key the early diagnosis, since this it would preserving the normal biomechanics of the spine. By this, many important is the adequate management in the Emergency Department. It should never be underestimated the problem, is necessary to have a good image to confirm it.”

Finally I would like to point out a small doubt about the linguistic rendering of the sentence present in line 59-60 "The time between injury and reduction is key, as it is directly related to the capacity to reduce the joint and the prognosis.". What is meant by "the capacity tu reduce the joint"?

ANSWER

The sentence is has been corrected as follow:

“The time between injury and reduction is key, as it is directly related to the capacity (facility) to reduce the joint and the prognosis.".

Reviewer 2 Report

Authors present a narrative review of C1/2 rotatory subluxation in adults. Unfortunately the authors did not include Materials and methods where they describe what was their methodology in search of papers and wheather PRISMA criteria was used. Also although this is narrative review, I suggest to include a table with most important manuscripts on the subject and include a Discussion section. There are unfortunately no MRI images of this rare disease, so I suggest to include. Furthermore, complications of the treatment are not thoroughly discussed. 

Several important references are missing and need to be included and thoroughly discussed:

Kia C, Mallozzi S, Moss I. Chronic Atlantoaxial Rotatory Subluxation in an Adult Following a Traumatic Event: A Case Report. Int J Spine Surg. 2020;14(4):488-492. doi:10.14444/7064

Yeung CY, Feng CK. Halter Traction for the Treatment of Atlantoaxial Rotatory Fixation. J Bone Joint Surg Am. 2022 Feb 2;104(3):229-238. doi: 10.2106/JBJS.21.00831. PMID: 34932516.

Tuan SH, Sun SF, Huang WY, Chen GB, Li MH, Liou IH. Effect of high intensity laser therapy in the treatment of acute atlantoaxial rotatory subluxation: A case report. J Back Musculoskelet Rehabil. 2022 Jan 14. doi: 10.3233/BMR-210133. Epub ahead of print. PMID: 35068439.

Horsfall HL, Gharooni AA, Al-Mousa A, Shtaya A, Pereira E. Traumatic atlantoaxial rotatory subluxation in adults - A case report and literature review. Surg Neurol Int. 2020 Nov 6;11:376. doi: 10.25259/SNI_671_2020. PMID: 33408910; PMCID: PMC7771491.

Ng C, Dominguez JF, Feldstein E, Houten JK, Spirollari E, Gandhi CD, Cole CD, Kinon MD. Does alar ligament injury predict conservative treatment failure of atlantoaxial rotatory subluxation in adults: Case report and review of the literature. Spinal Cord Ser Cases. 2021 Dec 3;7(1):103. doi: 10.1038/s41394-021-00464-9. PMID: 34862363; PMCID: PMC8642488.

The manuscript needs furthermore English language editing, either by a native speaker or editing service. 

Author Response

REVIEWER-2

Authors present a narrative review of C1/2 rotatory subluxation in adults. Unfortunately the authors did not include Materials and methods where they describe what was their methodology in search of papers and wheather PRISMA criteria was used. Also although this is narrative review, I suggest to include a table with most important manuscripts on the subject and include a Discussion section. There are unfortunately no MRI images of this rare disease, so I suggest to include. Furthermore, complications of the treatment are not thoroughly discussed. 

We thank you for the opportunity to revise the manuscript. We appreciate the careful review and constructive suggestions. We strongly think that the manuscript has been substantially improved after making the suggested changes  signaled in red into the text.

ANSWERS

First, following your suggestions, we have restructured the manuscript and defined the different aspects. 

Throughout the manuscript you can see in red the changes and extensions made, in the different sections:

- We have highlighted the objectives section 

- We have added and modified the material and methods section. In this section we have included the figure that you have suggested

- We have developed another new point (discussion) in which you will see the suggested table. Based on this and on what we have written above, we have developed this aspect of the discussion.

- We have included quotes that you have suggested, in different sections, in addition to others that have been necessary for the development of the discussion.

- We have added a paragraph in the discussion as also suggested by the other reviewer.

Round 2

Reviewer 2 Report

The authors have sufficiently responded to reviewers remarks.